# Somatosensory-Evoked Early Sharp Waves in the Neonatal Rat Hippocampus

**DOI:** 10.3390/ijms24108721

**Published:** 2023-05-13

**Authors:** Azat Gainutdinov, Dmitrii Shipkov, Mikhail Sintsov, Lorenzo Fabrizi, Azat Nasretdinov, Roustem Khazipov, Guzel Valeeva

**Affiliations:** 1Institut de Neurobiologie de la Méditerranée (INMED U1249), Aix-Marseille University, 13273 Marseille, France; azat.gainutdinov@inserm.fr; 2Laboratory of Neurobiology, Kazan Federal University, 420008 Kazan, Russiamikhail.yu.sintsov@gmail.com (M.S.); l.fabrizi@ucl.ac.uk (L.F.); gurvaleeva@kpfu.ru (G.V.); 3Department of Neuroscience, Physiology and Pharmacology, University College London, London WC1E 6BT, UK

**Keywords:** hippocampus, CA1, dentate gyrus, sharp wave, neonatal rat, somatosensory, sensory-evoked response, local field potentials, multiple unit activity, current-source density

## Abstract

The developing entorhinal–hippocampal system is embedded within a large-scale bottom-up network, where spontaneous myoclonic movements, presumably via somatosensory feedback, trigger hippocampal early sharp waves (eSPWs). The hypothesis, that somatosensory feedback links myoclonic movements with eSPWs, implies that direct somatosensory stimulation should also be capable of evoking eSPWs. In this study, we examined hippocampal responses to electrical stimulation of the somatosensory periphery in urethane-anesthetized, immobilized neonatal rat pups using silicone probe recordings. We found that somatosensory stimulation in ~33% of the trials evoked local field potential (LFP) and multiple unit activity (MUA) responses identical to spontaneous eSPWs. The somatosensory-evoked eSPWs were delayed from the stimulus, on average, by 188 ms. Both spontaneous and somatosensory-evoked eSPWs (i) had similar amplitude of ~0.5 mV and half-duration of ~40 ms, (ii) had similar current-source density (CSD) profiles, with current sinks in CA1 strata radiatum, lacunosum-moleculare and DG molecular layer and (iii) were associated with MUA increase in CA1 and DG. Our results indicate that eSPWs can be triggered by direct somatosensory stimulations and support the hypothesis that sensory feedback from movements is involved in the association of eSPWs with myoclonic movements in neonatal rats.

## 1. Introduction

Early sharp waves (eSPWs) are the earliest organized activity pattern in the developing hippocampus of neonatal rodents [1,2,3,4,5,6]. Electrophysiological traits of eSPWs are similar to those of adult SPWs, but their generative network mechanisms differ. While adult SPW-Rs are spontaneous top-down events, self-generated in the hippocampal circuit [7], eSPWs are primarily bottom-up events involving the inputs from the entorhinal cortex (EC). These inputs are activated during myoclonic movements (startles and twitches) characteristic of active sleep during the neonatal period in rodents [3,6,8,9,10]. Yet, network mechanisms linking eSPWs with myoclonic movements remain largely unknown. The current hypothesis involves somatosensory (proprioceptive and tactile) feedback generated by myoclonic movements [11,12], which trigger activity bursts in the somatosensory pathways [13,14,15,16,17,18,19,20]) that are further conveyed via the EC to the hippocampus [3,4,5,6,9,21]. The hypothesis that somatosensory feedback binds myoclonic movements with eSPWs implies that direct somatosensory stimulation should also be capable of evoking eSPWs. The hippocampus is a sensory hub that integrates sensory inputs of all modalities to enable its functions in spatial navigation, learning and memory [22]. In adults, somatosensory, auditory and visual stimuli evoke local field potential (LFP) responses in the hippocampus and modulate hippocampal neuron firing [23,24,25,26,27,28]. The current-source density (CSD) profile of somatosensory-evoked LFP responses reveals activation of temporoammonic and perforant pathways with characteristic sinks in the CA1 stratum lacunosum-moleculare and molecular layer of dentate gyrus, respectively [23,24]. These responses are also associated with the activation of granular cells in the dentate gyrus (DG) and variable but predominantly inhibitory effects on CA1 pyramidal neurons [23]. However, somatosensory-evoked responses in the hippocampus of neonatal rats remain largely unexplored.

In the present study, we examined the responses evoked by somatosensory stimulation in the hippocampus of neonatal rat pups (P5–P6) using silicone probe recordings along the CA1–DG axis. Experiments were conducted under urethane anesthesia to suppress movements. We found that in neonatal rats, somatosensory stimulation evoked ample hippocampal responses with CSD profiles suggesting involvement of temporoammonic and perforant inputs from the EC, as well as Schaffer collateral input from CA3, and global activation of CA1 and DG neurons. The electrographic portrait of somatosensory-evoked hippocampal responses matched that of spontaneous eSPWs. These findings support the hypothesis that sensory feedback from movements is involved in eSPW binding to myoclonic movements in neonatal rats and are compatible with delayed development of inhibition in the EC–hippocampal circuit.

## 2. Results

We explored somatosensory-evoked responses using 16-channel silicone probe recordings of local field potential (LFP) and multiple unit activity (MUA) from the dorsal hippocampus of P5-P6 Wistar rat pups, in which eSPWs represent the earliest and most prominent hippocampal activity pattern [2,5,6,29] (Figure 1A). The location of the recording sites was identified during post hoc analysis of the DiI electrode tracks in coronal brain sections (Figure 1B). Somatosensory responses were evoked by subcutaneous electrical stimulation of the contralateral whisker pad. Body movements were completely suppressed by urethane to isolate the “direct” somatosensory response uncontaminated by possible feedback from movements (Figure 1C). As illustrated by the example recording in Figure 1C (see also Figure 1D), the whisker pad stimulation evoked hippocampal LFP responses with a maximal negativity in CA1 stratum lacunosum-moleculare and polarity reversal near CA1 pyramidal cell layer/proximal stratum radiatum, as well as increased firing of CA1 and DG units.

These somatosensory-evoked responses were almost identical to spontaneous eSPWs in their spatiotemporal characteristics (Figure 2) and MUA modulation (see below) and therefore will be referred to hereafter as somatosensory-evoked eSPWs, or evoked eSPWs. The somatosensory-evoked eSPW peak was delayed from the stimulus, on average, by [median (first quartile–third quartile)] 188 (178–211) ms (*n* = 9 rats; Figure 2A,B), with a delay jitter of 25 (20–32) ms. Spontaneous eSPWs occurred with an average frequency of 0.019 (0.014–0.029) s^−1^ (*n* = 9 rats), and the probability of eSPW generation calculated within 500 ms time window before and after the stimulus reached 1.7 (0.8–2.6)% and 32.7 (21.2–51.7)%, respectively (*n* = 9 rats; *p* = 0.004; Figure 2A,C). Somatosensory-evoked eSPWs had an average amplitude of 573 (455–601) μV and did not differ significantly from the spontaneous eSPW amplitude of 497 (440–650) μV (*n* = 9 rats; *p* = 1; Figure 2E,F). Spontaneous and evoked eSPWs also had similar half-widths, reaching 41 (36–48) ms and 43 (39–52) ms, respectively (*n* = 9 rats; *p* = 0.098; Figure 2D).

The current-source density analysis also revealed CSD profiles of somatosensory-evoked eSPWs matching those of spontaneous eSPWs (Figure 3A). For a detailed description of eSPW CSD profiles, we selected six of the nine recorded animals where the probe sites completely covered the CA1 str. oriens—DG str. granulosum distance. Sink 1 in CA1 str. radiatum, reflecting the activation of CA3 ⇒ CA1 Schaffer collateral synapses, was located at an average depth of 48 (0–75) μm from the CA1 pyramidal cell layer during both sensory-evoked and spontaneous eSPWs (*n* = 6 rats; *p* = 1.0; Figure 3A,B). Another major current Sink 2, previously identified as a result of EC input activation [3,6], was located within str. lacunosum-moleculare, 241 (225–251) μm below the CA1 pyramidal cell layer during both sensory-evoked and spontaneous eSPWs (*n* = 6 rats; *p* = 1.0; Figure 3A,B). During both types of eSPWs, we also observed a third current sink localized in the external two-thirds of the DG str. moleculare (Sink 3). The average distance of Sink 3 from the hippocampal fissure was 50 (35–77) μm both for sensory-evoked and spontaneous eSPWs (*n* = 5 rats; *p* = 1.0; Figure 3A,B). Figure 3B represents the relative location of all three current sinks normalized to CA1 pyramidal cell layer—fissure distance in six individual animals.

The comparison of current sink amplitudes revealed that Sink 2 was the most prominent of all three sinks observed during both spontaneous and sensory-evoked eSPWs. The magnitude of Sinks 1 and 3 reached 0.39 (0.29–0.48) and 0.29 (0.24–0.48) of Sink 2 for spontaneous eSPWs and 0.42 (0.31–0.48) and 0.32 (0.21–0.50) of Sink 2 for sensory-evoked eSPWs, respectively (*n* = 6 rats; Figure 4).

We also found that Sink 2 occurred earlier than Sink 1 and Sink 3 (Figure 5). The current sink times were defined as the time of sink peak amplitude. During spontaneous eSPWs, Sink 1 and Sink 3 were delayed by 2.0 (1.0–7.0) ms and 17.0 (9.5–40.3) ms in relation to the negativity peak of eSPWs, while Sink 2 appeared 3.5 (3.0–4.0) ms earlier than the eSPW peak (*n* = 6 rats; Figure 5A). The current sink times of spontaneous eSPWs were significantly different from each other, as shown by the *p*-value map in Figure 5B. The temporal order of the main current sinks was identical for sensory-evoked eSPWs (*n* = 6 rats; Figure 5A). Both Sink 1 and Sink 3 of sensory-evoked eSPWs were observed later than eSPW peak time with delay values reaching 8.0 (0.0–9.0) ms and 12.0 (6.8–43.3) ms, respectively (*n* = 6 rats, Figure 5A,B). At the same time, Sink 2 preceded eSPW peak by 4.0 (2.0–5.0) ms (*n* = 6 rats; Figure 5A,B). Although Sinks 1 and 3 were delayed from Sink 2, their timing was not significantly different from each other (Figure 5B). Taken together, the similarities in CSD profiles described above suggest the common network mechanisms underlying spontaneous and somatosensory-evoked eSPW generation.

The eSPWs generated in response to stimulation were associated with an increase in MUA in CA1 pyramidal cell layer and DG granular layer, also characteristic of spontaneous eSPWs (Figure 6). In four of the five recorded rat pups, CA1 pyramidal cell neurons fired before DG granular layer neurons during eSPWs. The averaged MUA z-score peak time was observed, on average, at 0 (−5–0) ms in relation to the eSPW peak in CA1 pyramidal cell layer and 30 (20–55) ms after the eSPW peak in the DG granular layer during spontaneous eSPWs (*n* = 4 rats; *p* = 0.029; Figure 6C). During sensory-evoked eSPWs, MUA z-score peaked at −5 (−10–5) ms before the eSPW peak in CA1 pyramidal layer and 45 (25–65) ms after the eSPW peak in DG granular layer (*n* = 4 rats; *p* = 0.029; Figure 6C). In one animal, MUA peak times coincided for the pyramidal and granular layers during spontaneous eSPWs, and the granular layer MUA peak appeared earlier than peak neuronal firing in pyramidal cell layer.

## 3. Discussion

In the present study, we explored somatosensory-evoked responses in the hippocampus of neonatal rats after suppression of motor startle responses under urethane anesthesia. We found that somatosensory stimulation triggers ample hippocampal responses with spatiotemporal characteristics identical to spontaneous eSPWs, including CSD profiles, suggesting involvement of temporoammonic and perforant inputs from EC and Schaffer collateral input from CA3 as well as global activation of CA1 and DG neurons. These findings support the hypothesis that sensory feedback from movements is involved in eSPW binding to myoclonic movements in neonatal rats. The results of our study also highlight an important developmental difference in somatosensory-evoked hippocampal responses, likely reflecting the delayed development of inhibitory hippocampal circuits.

Previous studies hypothesized that binding of eSPWs with myoclonic movements involves reafferentation by sensory (tactile and proprioceptive) feedback activated during spontaneous myoclonic movements, which occur in the neonatal rodents during active sleep [1,4,6,21]. Neonatal myoclonic movements have been previously shown to evoke bottom-up neuronal activation along the somatosensory pathways, including the dorsal layers of the spinal cord, relay thalamus, primary somatosensory cortex and hippocampus [11,13,15,21,30,31,32,33]. However, whether somatosensory signals can reach the hippocampus remained hypothetical. In the present study, we found that direct somatosensory stimulation reliably evoked hippocampal eSPWs even after suppression of motor startle responses under urethane anesthesia. These findings directly indicate that somatosensory signals indeed can reach the neonatal hippocampus, thus supporting the network model of eSPWs in which sensory feedback from movements triggers hippocampal eSPWs. Importantly, sensory input is not an obligatory condition for the occurrence of eSPWs. In fact, eSPWs may arise spontaneously without any accompanying movements in behaving pups [6], as well as during quiet sleep-like state in immobilized pups under urethane anesthesia (Ref. [2] and the present study). Moreover, eSPWs become less frequent and dissociate from myoclonic movements but persist in the “cerveau isole” preparation following a supracollicular transection that severs external inputs [8]. Hence, eSPWs are events that are generated internally, and sensory input only serves as a trigger for their occurrence.

CSD analysis revealed similarities in the generative mechanisms of spontaneous and somatosensory-evoked eSPWs, including an involvement of extrinsic (from EC to CA1 and DG) and intrahippocampal (from CA3 to CA1) inputs in eSPW generation. While eSPWs’ Sink 1 in CA1 *str. radiatum* (presumably generated by CA3–CA1 Schaffer collateral synapses) was consistent with previous studies [3,6], the previously described eSPWs’ Sink 2 near the fissure was actually composed of two sinks clearly separated by the hippocampal fissure: Sink 2 in CA1 *str. lacunosum-moleculare* (presumably generated by EC L3 ⇒ CA1 synapses on distal dendrites of CA1 pyramidal cells) and Sink 3 in DG *str. moleculare* (presumably generated by EC L2 ⇒ DG synapses on dendrites of DG granular cells). These findings align with co-activation of EC L2 and L3 neurons during population bursts, which are generated in EC following myoclonic movements and precede hippocampal eSPWs [6,34]. Additionally, in this study, DG Sink 3 was delayed from Sink 2 in CA1 *str. lacunosum-moleculare*, along with delayed activation of DG units from CA1 units, which differs from the more synchronous DG and CA1 neuronal discharge during eSPWs reported previously [6]. This suggests that EC layer 2 discharge is delayed from L3 during EC population bursts driving eSPWs, possibly due to the use of urethane in the present study. The delayed activation of the perforant pathway from the temporoammonic pathway may also explain delayed discharge of DG-driven CA3 neurons, as reflected by the delayed CA1 *str. radiatum* Sink 1 from str. *lacunosum-moleculare* Sink 2. Regardless of these differences, there was a good correspondence in CSD profiles and neuronal activation in CA1 and DG during both spontaneous and sensory-evoked eSPWs.

Somatosensory-evoked eSPWs described in the present study share many common features with somatosensory-evoked hippocampal responses in adult animals but also display important developmental differences. In adults, sensory stimulation in different modalities evokes LFP responses and modulates neuronal activity both in the hippocampus and DG [23,24,25,26,27,28]. The most prominent sinks of sensory-evoked responses are located in DG *str. moleculare* along with a less consistent sink in CA1 *str. lacunosum-moleculare*, which correspond to Sinks 3 and 2 of the sensory-evoked eSPWs of the present study, respectively [23,24]. This suggests that sensory-evoked responses involve activation of EC layers 2 and 3 both in neonates and adults. However, sensory responses in adults lack the sink in CA1 *str. radiatum*, which is characteristic of eSPWs in neonates, suggesting that CA3 ⇒ CA1 Schaffer collateral synapses are not activated during sensory responses in adults. This is in line with the global inhibition exerted by perforant input on CA3 neurons [35]. Additionally, neuronal activation during sensory responses is observed in adults as well as in neonates in DG [23,24]. Yet, CA1 units show variable change in firing with predominantly inhibitory response to sensory stimuli, which is associated with GABAergic hyperpolarization in the majority of CA1 pyramidal neurons [23]. The mechanisms underlying developmental transition of somatosensory-evoked responses from eSPWs, which are generated by feedforward excitation in entorhinal–hippocampal and intrahippocampal circuits, to more sparse and variable responses in adults remain unknown. A likely candidate is the delayed development of hippocampal inhibitory circuits [10,36,37,38,39,40,41,42], which may explain the lack of CA3 activation in adults as a result of feedforward CA3 inhibition by inputs from DG and inhibition of the majority of CA1 units during sensory-evoked response.

In conclusion, results of the present study support the hypothesis that somatosensory feedback is involved in binding hippocampal eSPWs with myoclonic movements in neonatal rats. By providing sequential activation of all major synaptic pathways and “causality” in the timing of activation of pre- and postsynaptic elements within the large-scale bottom-up circuit processing somatosensory information, such mode of function, supported by spontaneous movements, may serve for the activity-dependent formation of these circuits [1,43,44,45,46,47,48].

## 4. Materials and Methods

Wistar rats of either sex from postnatal days (P) 5–6 were used. Preparation of the animals for recordings was performed under isoflurane anesthesia. Recordings were performed from head-restrained urethane-anesthetized rats (1.3–1.5 g/kg). A metal ring was fixed to the skull with a self-curing acrylic denture repair material (Meliodent RR, Kulzer, GmbH, Hanau, Germany) and via ball-joint to a magnetic stand. Animals were surrounded by a cotton nest and heated via a thermal pad (35–37 °C) throughout the recording session.

Extracellular recordings of local field potentials (LFPs) and multiple unit activity (MUA) were performed along the CA1–dentate gyrus axis of left dorsal hippocampus using a 16-site linear silicon probes with 50 μm separation distance between the electrodes (NeuroNexus, Ann Arbor, MI, USA). DiI-coated silicone probes were placed using stereotaxic coordinates [49]. A chlorided silver wire inserted into occipital or frontal cortex served as a ground electrode. Signals from extracellular recordings were amplified and filtered (×10,000; 0.15–4 kHz) using a Digital Lynx SX amplifier (Neuralynx, Bozeman, MT, USA) and digitized at 32 kHz. One recording session lasted from 30 min to an hour. Somatosensory stimulation was applied to the right whiskerpad, forelimb or hindlimb via a pair of stainless-steel subcutaneous needle electrodes. Square current pulses of 4–7 mA (50 μs duration) were generated by a Master-8 stimulator (A.M.P.I., Jerusalem, Israel) with a 10–20 s interpulse interval. Body movement recordings were performed using piezoelectric sensors attached to both forelimbs and a hindlimb.

After recordings, the animals were deeply anesthetized with urethane (3 g/kg, intraperitoneally). The brains were removed and left for fixation in 4% paraformaldehyde and 1% glutaraldehyde (Sigma-Aldrich, Steinheim, Germany) solution for a few days. Next, 100 μm-thick coronal slices were cut using a Vibratome (Thermo Fisher Scientific, Waltham, MA, USA). Probe positions were identified from the DiI tracks overlaid on the microphotographs of hippocampal sections after cresyl violet staining. The recording site locations was verified by the highest MUA rate in CA1 *str. pyramidale* and LFP polarity reversion observed during eSPWs across the CA1 *str. pyramidale/str. radiatum* border.

Electrophysiological data processing and analysis were performed as described previously [6,15,30]. In brief, wideband recordings were preprocessed using custom-written functions in MATLAB (MathWorks, Natick, MA, USA). eSPWs were detected semi-automatically from down-sampled (1000 Hz), bandpass filtered (1–100 Hz, Chebyshev Type 2 Filter) LFP signals. First, the filtered LFP signal from the channel located in CA1 pyramidal cell layer (positive LFP signal during eSPWs) was subtracted from the LFP trace recorded from the channel in *str. lacunosum-moleculare* (the most negative LFP signal during eSPWs). In the resulting trace, all events reaching an amplitude greater than 3 standard deviations were considered as putative eSPWs. Afterwards, LFP segments of the original 16-channel recording from −1 s to 1 around the putative eSPW were visually inspected to select eSPWs. eSPW amplitude and half-width was estimated using the trace resulting from LFP subtraction between channels located in *str. lacunosum-moleculare* and *str. pyramidale*.

To eliminate volume conductance and localize synaptic currents, CSD analysis across depth was performed on LFP traces normalized to the maximum amplitude signal across all channels and averaged across events. CSD was computed for each recording site according to a differential scheme for second derivative and smoothed with a triangular kernel of length 3 [50]. For MUA analysis, we considered five animals showing well-detectable and stable neuronal firing in pyramidal and granular layers throughout the length of recording session. For MUA detection, raw data were filtered using a 200–1000 Hz bandpass filter and spikes were detected as negative events exceeding 1.8–3 standard deviations (STD) of the filtered signal. STD was calculated individually for each of 16 recording channels. Normalized MUA z-scores were calculated as z-score = (x–μ)/σ, where x—MUA PETH value, μ—mean value and σ—standard deviation.

Statistical analysis was performed using the MATLAB Statistics toolbox. Statistical comparisons were carried out using paired-sample Wilcoxon signed-rank and Mann–Whitney tests. A *p*-value of less than 0.05 was considered significant. Group data are presented as median (first quartile–third quartile).

## Figures and Tables

**Figure 1 ijms-24-08721-f001:**
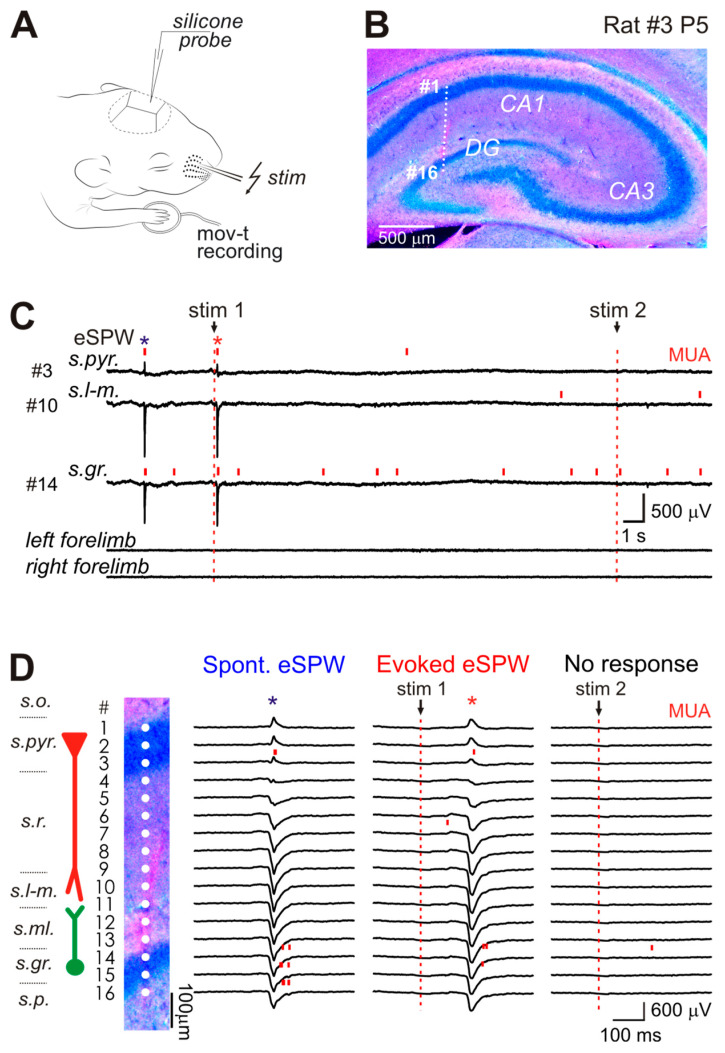
Spontaneous early sharp waves (eSPWs) and hippocampal responses evoked by electrical stimulation of skin in neonatal rat pup. (**A**), Experimental setup represents silicone probe recording from left hippocampus of P5–P6 (P—postnatal day) rat pups during subcutaneous electrical stimulation of the right whisker pad. (**B**), Post hoc microphotograph of 100 μm-thick coronal hippocampal slice (cresyl violet staining) overlaid with recording electrodes of 16-channel silicone probe. (**C**), Hippocampal activity recorded from P5 rat on the probe sites shown in panels B and D. Spontaneous eSPWs and stimulus-evoked hippocampal responses are marked by blue and red asterisks (*), respectively. Simultaneous limb movement recordings from two piezo-detectors are shown below. (**D**), The reconstruction of recoding electrode locations across hippocampal layers (s.o., CA1 str. oriens; s.pyr., str. pyramidale; s.r., str. radiatum; s.l-m., str. lacunosum-moleculare; s.ml., DG str. moleculae; s.gr., DG str. granulosum; s.p., DG polymorphic cell layer) and examples of individual spontaneous hippocampal eSPW (left), individual hippocampal response evoked by electrical stimulus (middle) and the stimulus evoking no response in the hippocampus (right). Red bars show multiple unit activity (MUA). The vertical dashed lines correspond to stimulus time.

**Figure 2 ijms-24-08721-f002:**
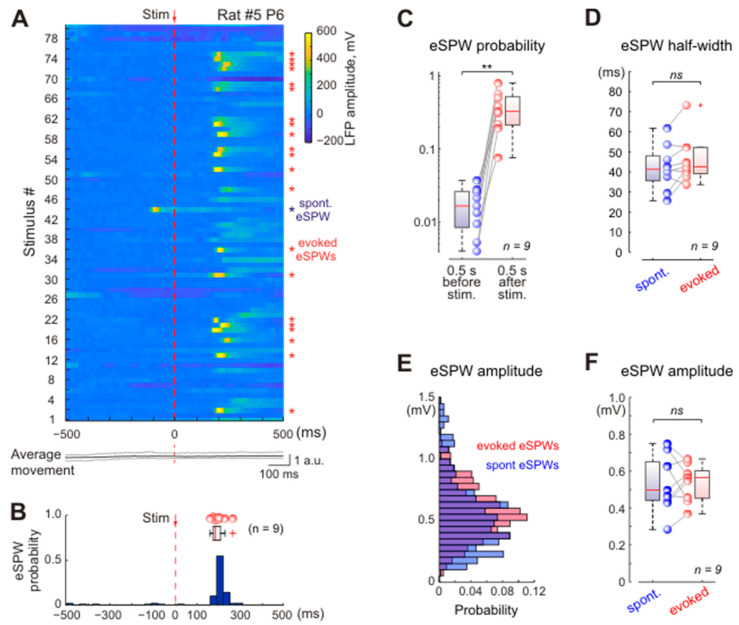
Somatosensory-evoked hippocampal eSPW response properties. (**A**), Stimulus-triggered raster plot of hippocampal local field potential (LFP) responses evoked by the first 80 stimuli in a P6 rat. The stimuli that evoked eSPW responses are marked by red asterisks (*) on the right, blue asterisk indicates a spontaneous eSPW. Below, corresponding movement trace averaged over three piezo-channels (black) with a 25–75% interquartile range (gray). Dashed red line shows the stimulus time (0 ms). (**B**), PSTH of all eSPW responses recorded from the rat shown in (**A**). A boxplot above the histogram shows group data on eSPW response peak time in nine P5-P6 rats. Circles—median peak time values in individual animals. Outliers are shown by red crosses. (**C**–**F**), Group data on probability (**C**), half-width (**D**) and amplitude (**E**,**F**) of spontaneous eSPWs and evoked eSPW responses in nine P5-P6 rats. Circles show corresponding median values from individual animals. **, *p*-value < 0.01; *ns*, not significant.

**Figure 3 ijms-24-08721-f003:**
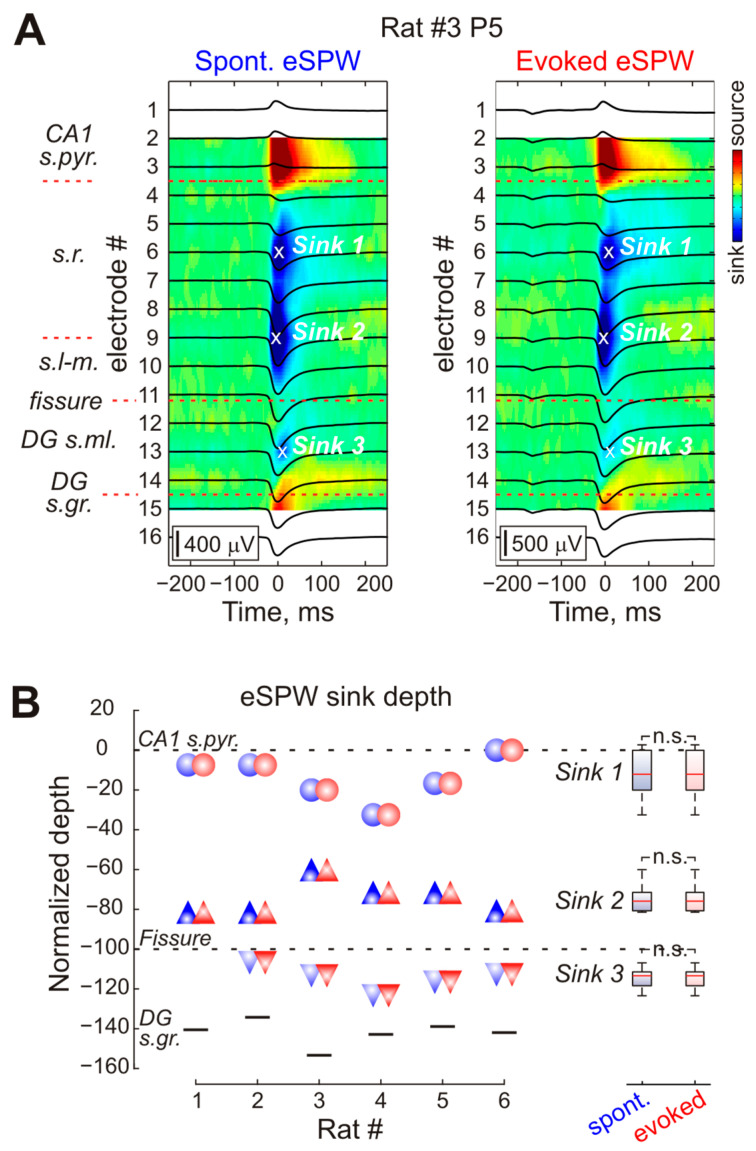
Current-source density (CSD) profile of spontaneous and stimulus-evoked eSPWs. (**A**), Average LFP traces recorded in P5 rat during spontaneous and stimulus-evoked eSPWs overlaid on corresponding CSD maps. Hippocampal layer borders are shown on the left (CA1 s.pyr.—str. pyramidale, s.r.—str. radiatum, s.l-m.—str. lacunosum-moleculare, DG s.ml.—str. moleculare, DG s.gr.—str. granulosum). (**B**), Group data on the location of three main current sinks (Sink 1—circles, Sink 2—triangles, Sink 3—inverted triangles) in CSD profiles of spontaneous (blue) and stimulus-evoked (red) eSPWs. Sink depths are normalized to CA1 s.pyr./s.r.–fissure distance. Boxplots on the right represent statistical data on sink depths. n.s.—not significant.

**Figure 4 ijms-24-08721-f004:**
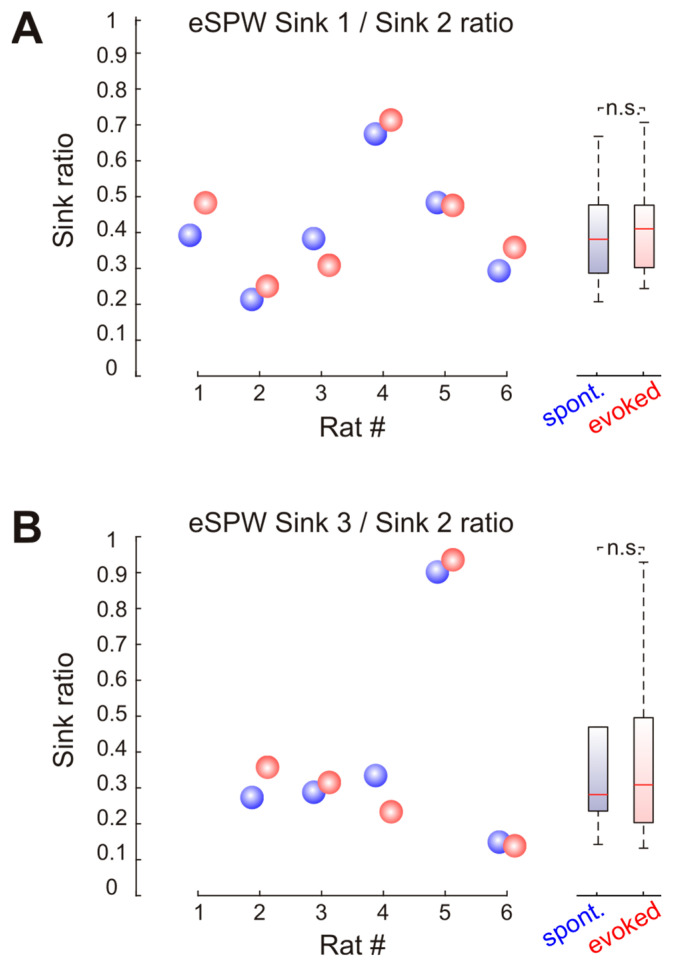
The relative magnitude of spontaneous and stimulus-evoked eSPWs. Group data on Sink 1/Sink 2 (**A**) and Sink 3/Sink 2 (**B**) amplitude ratio. Boxplots on the right show statistical data for spontaneous (blue) and somatosensory-evoked (red) eSPWs. Circles show median sink ratio values for spontaneous (blue) and stimulus-evoked (red) eSPWs in six individual P5–P6 rats. n.s.—not significant.

**Figure 5 ijms-24-08721-f005:**
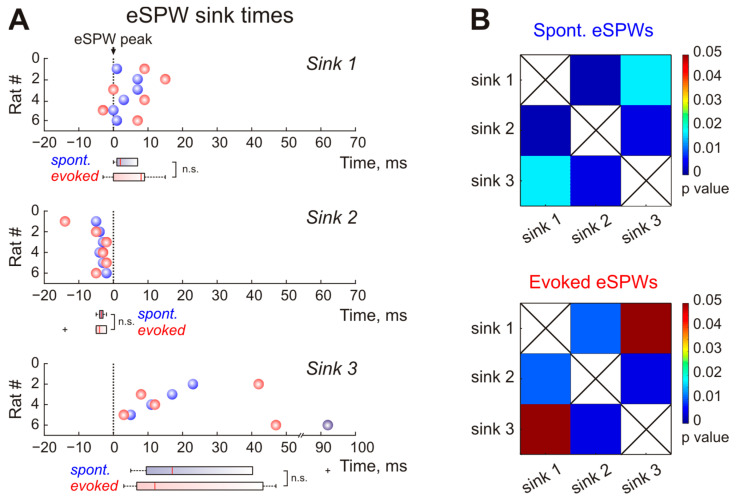
Time relations between three main current sink observed during spontaneous and stimulus-evoked eSPWs. (**A**), Group data on Sink 1, Sink 2 and Sink 3 times of spontaneous (blue circles) and stimulus-evoked (red circles) eSPWs in relation to eSPW peak (t = 0 ms). Corresponding boxplots are shown below each plot. n.s.—not significant. (**B**), *p*-value maps for current sink times of spontaneous (**top**) and stimulus-evoked (**bottom**) eSPWs.

**Figure 6 ijms-24-08721-f006:**
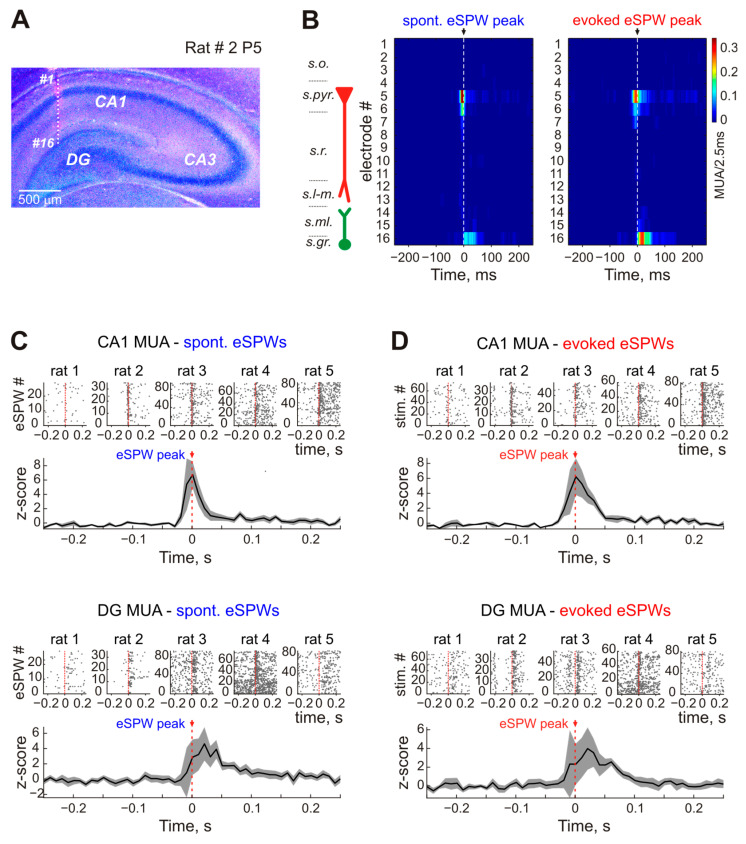
Multiple unit activity (MUA) recorded in CA1 pyramidal cell layer and DG granular cell layer during spontaneous and stimulus-evoked eSPW. (**A**), Post hoc reconstruction of 16 recording electrode locations across the hippocampal layers of a P5 rat. (**B**), MUA density plots for spontaneous eSPWs and stimulus-evoked eSPW responses. Recording site numbers correspond to those shown in panel A. Hippocampal layer borders are shown on the left—*s.o.*, CA1 str. oriens; *s.pyr.*, str. pyramidale; *s.r.*, str. radiatum; *s.l-m.*, str. lacunosum-moleculare; *s.ml.*, DG str. moleculare; *s.gr.*, DG str. granulosum. (**C**,**D**), Raster plots and average PETHs for MUA detected in CA1 s. pyr. (**top**) and DG s. gr. (**bottom**) during spontaneous eSPWs (**C**) and stimulus-evoked eSPW responses (**D**).

## Data Availability

Original and processed data and signal processing and analysis routines are available on request from the authors.

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
