# Peer review of "Somatosensory-Evoked Early Sharp Waves in the Neonatal Rat Hippocampus"

_ijms, 2023, doi:10.3390/ijms24108721_

Round 1
Reviewer 1 Report
The study by Gainutdinov, Shipkov et al. examines the role of somatosensory input for the generation of early sharp waves (eSPWs) in the developing hippocampus in vivo. The authors test a hypothesis derived from earlier work stating that sensory feedback from spontaneous myoclonic body movements represents a major drive for early hippocampal activity. The question under investigation is timely, and the main finding is of high relevance for a systems physiology perspective on hippocampal maturation.
The ms is very well written. My overall impression is that experiments and analyses were performed with great care. The discussion is clear and balanced. I only have a few minor points that could be considered to further improve the ms:
(1) The use of urethane is well justified. However, as urethane completely suppressed body movements, the brain state under urethane differs from brain states in unanesthetized pups (it is more reminiscent of quiet sleep). Since the association of myoclonic movements to hippocampal activity was previously reported to be state-dependent, this aspect should be explicitly discussed.
(2) p. 6, l. 157-158 (“While each one …”): Please rephrase the sentence for clarity.
(3) p. 9, l. 235: “form” should read “from”.
(4) p. 10, l. 253 (“This is in line … “): A reference for this statement is required.
(5) Methods: Please provide details for the piezo-based recording of body movements.
Author Response
We would like to thank the reviewer for his/her thoughtful comments, which have been addressed in the revised manuscript. Please find below point-by-point response to the comments.
(1) The use of urethane is well justified. However, as urethane completely suppressed body movements, the brain state under urethane differs from brain states in unanesthetized pups (it is more reminiscent of quiet sleep). Since the association of myoclonic movements to hippocampal activity was previously reported to be state-dependent, this aspect should be explicitly discussed.
RESPONSE:
We thank the reviewer for pointing out this important aspect of our study. We address this issue in the Discussion section, in the paragraph on lines 219-237, which has been modified to respond to this comment as follows:
“Previous studies hypothesized that binding of eSPWs with myoclonic movements involves reafferentation by sensory (tactile and proprioceptive) feedback activated during spontaneous myoclonic movements, which occur in the neonatal rodents during active sleep [1, 4, 6, 17]. Neonatal myoclonic movements have been previously shown to evoke bottom-up neuronal activation along the somatosensory pathways, including the dorsal layers of the spinal cord, relay thalamus, primary somatosensory cortex and hippocampus [11, 13, 15, 17, 25-28]. However, whether somatosensory signals can reach the hippocampus remained hypothetical. In the present study, we found that direct somatosensory stimulation reliably evokeв hippocampal eSPWs even after suppression of motor startle responses under urethane anesthesia. These findings directly indicate that somatosensory signals indeed can reach the neonatal hippocampus, thus supporting the network model of eSPWs in which sensory feedback from movements triggers hippocampal eSPWs. Importantly, sensory input is not an obligatory condition for the occurrence of eSPWs. In fact, eSPWs may arise spontaneously without any accompanying movements in behaving pups [6], as well as during quiet sleep-like state in immobilized pups under urethane anesthesia (Ref. [2] and the present study). Moreover, eSPWs become less frequent and dissociate from myoclonic movements, but persist in the “cerveau isole” preparation following a supracollicular transection that severs external inputs [8]. Hence, eSPWs are events that are generated internally, and sensory input only serves as a trigger for their occurrence.”
(2) p. 6, l. 157-158 (“While each one …”): Please rephrase the sentence for clarity.
RESPONSE:
This has been rephrased as follows (Lines 174-175):
Although Sinks 1 and 3 were delayed from Sink 2, their timing was not significantly different from each other
(3) p. 9, l. 235: “form” should read “from”.
RESPONSE:
Corrected
(4) p. 10, l. 253 (“This is in line … “): A reference for this statement is required.
RESPONSE:
Reference [35] has been added (Line 271).
(5) Methods: Please provide details for the piezo-based recording of body movements.
RESPONSE:
This information is added to the revised manuscript as follows (Lines 307-308):
Body movement recordings were performed using piezoelectric sensors attached to both forelimbs and a hindlimb.
Reviewer 2 Report
Manuscripts were prepared, and experimental results were reported in the right direction. After improving these suggestions, the manuscript will be completed;
- Authors should give a brief idea about their significant experimental results in abstract section in mathematical-manner
- Authors need a diagram/graphical abstract to explain their pathway relation
- Authors should provide deeper background for experimental design (why the age of rats was selected, the reason for their strain and quantification methods)
- Authors should provide more references in their discussion and introduction sections.
Author Response
We would like to thank the reviewer for the valuable comments. Please find below our point-by-point response to the comments.
- Authors should give a brief idea about their significant experimental results in abstract section in mathematical-manner
RESPONSE:
In keeping with this recommendation, we have revised experimental results in abstract section as follows:
Lines 18-24
“We found that somatosensory stimulation in ~ 33% of the trials evoked local field potential (LFP) and multiple unit activity (MUA) responses identical to spontaneous eSPWs. The somatosensory-evoked eSPWs were delayed from the stimulus, on average, by 188 ms. Both spontaneous and somatosensory-evoked eSPWs: (i) had similar amplitude of ~ 0.5 mV and half-duration of ~ 40 ms, (ii) had similar current-source density (CSD) profiles, with current sinks in CA1 strata radiatum, lacunosum-moleculare, and DG molecular layer, and (iii) were associated with MUA increase in CA1 and DG.”
2) Authors need a diagram/graphical abstract to explain their pathway relation
RESPONSE:
We have prepared graphical abstract as recommended by the reviewer, and placed it below the text abstract in the revised manuscript.
3) Authors should provide deeper background for experimental design (why the age of rats was selected, the reason for their strain and quantification methods)
RESPONSE:
According to this comment, we provide the requested information in the revised manuscript as follows:
Age and strain: Lines 69-72
“We explored somatosensory-evoked responses using 16-channel silicone probe recordings of local field potential (LFP) and multiple unit activity (MUA) from the dorsal hippocampus of P5-P6 Wistar rat pups, in which eSPWs represent the earliest and most prominent hippocampal activity pattern [2, 5, 6, 25].”
Quantification methods: Lines 317-318
“Electrophysiological data processing and analysis were performed as described previously [6, 15, 26].”
4) Authors should provide more references in their discussion and introduction sections.
We have added 11 references in the introduction and discussion, increasing the total number of references from 39 to 50.